# Letermovir Primary Prophylaxis in High-Risk Hematopoietic Cell Transplant Recipients: A Matched Cohort Study

**DOI:** 10.3390/vaccines9040372

**Published:** 2021-04-12

**Authors:** Léna Royston, Eva Royston, Stavroula Masouridi-Levrat, Nathalie Vernaz, Yves Chalandon, Christian Van Delden, Dionysios Neofytos

**Affiliations:** 1Division of Infectious Diseases, University Hospital of Geneva, 1205 Geneva, Switzerland; lena.royston@hcuge.ch (L.R.); christian.vandelden@hcuge.ch (C.V.D.); 2Bone Marrow Transplant Unit, Division of Hematology, University Hospital of Geneva, 1205 Geneva, Switzerland; royston.eva@gmail.com (E.R.); stavroula.masouridi@hcuge.ch (S.M.-L.); yves.chalandon@hcuge.ch (Y.C.); 3Medical Directorate, University Hospital of Geneva, 1205 Geneva, Switzerland; nathalie.vernaz@hcuge.ch

**Keywords:** letermovir, cytomegalovirus (CMV), prophylaxis, allogeneic hematopoietic cell transplant recipients, recurrent CMV infection

## Abstract

Background: Real-life data on the administration of letermovir as cytomegalovirus (CMV) primary prophylaxis after allogeneic hematopoietic cell transplantation (HCT) remain limited. Methods: We conducted a retrospective single-center matched cohort study, comparing consecutive high-risk allogeneic HCT recipients (cases) receiving primary prophylaxis with letermovir and untreated matched historical controls, during a study period of 180 days. The primary outcome was the incidence of clinically significant (cs) CMV infection. Secondary outcomes included duration and costs of CMV-antiviral treatments, hospital resource utilization, hematology and laboratory parameters. Results: Letermovir prophylaxis decreased csCMV infection incidence from 82.7% (controls) to 34.5% (cases; *p*-value < 0.0001). Controls were more likely to have >1 episode of csCMV infection (59.6%) compared to cases (11.5%; *p*-value < 0.0001). Letermovir was associated with: shorter overall CMV-associated treatment duration (49 days vs. 77.8 days; *p*-value: 0.02) and a trend for lower costs of CMV-associated treatments ($4096 vs. $9736; *p*-value: 0.07) and reduced length of stay (44.8 days vs. 59.8 days; *p*-value: 0.16). Letermovir administration was associated with significantly shorter duration (27.3 days vs. 57.1 days; *p*-value: 0.008) and lower costs ($1089 vs. $2281; *p*-value: 0.008) of valganciclovir treatment. Compared to controls, higher platelet counts were observed in cases (138 G/L vs. 92 G/L; *p*-value: 0.03) and renal function was improved (94 mL/min/1.73 m^2^ vs. 74 mL/min/1.73 m^2^; *p*-value: 0.006). Conclusions: Primary anti-CMV letermovir prophylaxis decreased the incidence of csCMV infection and the administration of CMV-associated treatments and costs, particularly those associated with valganciclovir. An effect of letermovir on platelet reconstitution and renal function of csCMV post-HCT was observed and needs further investigation.

## 1. Background

Despite effective therapies, cytomegalovirus (CMV) reactivation remains a considerable threat after an allogeneic hematopoietic cell transplantation (HCT), with significant associated morbidity and costs [1,2,3,4,5]. Considering the toxicities associated with the until recently available anti-CMV agents, primary CMV prophylaxis is not routinely administered [6,7]. In contrast, a preemptive approach is usually followed, with regular measurements of blood CMV quantitative polymerase-chain reaction (qPCR) and treatment initiation based on documentation of CMV reactivation. Although preemptive treatment dramatically decreased CMV end-organ disease, CMV reactivation and preemptive treatment initiation induce significant toxicities, including myelotoxicity and nephrotoxicity, and are associated with prolonged length of stay (LOS) and higher inpatient costs [2,7,8,9,10]. Altogether, HCT-recipient CMV seropositive status remains associated with poor clinical outcomes after allogeneic HCT in the current era [9].

Letermovir, a CMV DNA terminase inhibitor targeting UL56, a subunit of the viral terminase complex, was recently approved as primary CMV prophylaxis during the first 14 weeks after allogeneic HCT [11]. Due to its efficacy and excellent safety profile, this agent offers a promising alternative to standard preemptive treatments, even though real-life data on letermovir prophylaxis remain limited [12,13,14]. We, therefore, performed a retrospective matched cohort study to describe the effect of letermovir on the incidence of clinically significant (cs) CMV infections, related morbidity, hospital resource utilization, and overall mortality in a real-life setting.

## 2. Methods

### 2.1. Design and Case-Control Matching

Cases included all consecutive adult (≥18-year-old) allogeneic HCT recipients who received primary CMV-prophylaxis with orally (PO) administered letermovir between 1 May 2019 and 31 May 2020. Starting 1 May 2019, PO letermovir primary CMV-prophylaxis is administered at our institution in: (i) CMV donor-negative (D-)/recipient-positive (R+) allogeneic HCT recipients from post-HCT-day 1 to post-HCT-day 100 and (ii) CMV R+ allogeneic HCT recipients with early (during the first 6 months post-HCT) grade ≥ 2 graft-versus-host disease (GvHD) requiring corticosteroid treatment at a dose ≥1 mg/kg/day and until tapering to <10 mg/day of prednisone equivalent.

Cases were matched 1:2 to consecutive patients transplanted from 1 January 2015 to 1 May 2019. Matching criteria consisted of the following variables: (i) underlying hematologic malignancy (myeloid/lymphoid), (ii) HCT donor type (HLA-matched related/matched or mismatched unrelated/haploidentical), (iii) conditioning regimen type (myeloablative/reduced intensity), and (iv) CMV D/R serology status. In addition, for those cases that received letermovir for early GvHD ≥ grade 2, cases and controls were matched for organ involvement (gastrointestinal tract/other). The study was approved by the local Ethics Committee. 

### 2.2. Study Hypothesis and Outcomes 

We hypothesized that administration of letermovir in high-risk allogeneic HCT recipients as primary prophylaxis is associated with decreased rates of csCMV infection, associated costs, in-hospital LOS, and all-cause mortality during the first 180 days post-study inclusion. For allogeneic CMV D-/R+ HCT recipients, the study period coincided with the first 180 days post-HCT. Patients included in the CMV R+/GvHD group were followed until day 180 post-diagnosis of GvHD. The primary outcome was the incidence of csCMV infection during the first 180 days post-study inclusion. The following secondary outcomes were studied: (i) >1 episode of csCMV infection and letermovir-resistance development, (ii) duration and costs of CMV-active antiviral treatment administration with (val)ganciclovir, foscarnet, and cidofovir for the treatment of csCMV infection, (iii) LOS, (iv) number of readmissions, (v) all-cause mortality, and (vi) rates of non-CMV double stranded (ds) DNA viral infections (all herpesviruses, adenovirus, BK-virus). Additional analyses were performed: (i) laboratory measurements, including absolute counts of neutrophils, lymphocytes, and platelets, and liver and renal function and (ii) hematology parameters, including relapse by day 180 after study inclusion and time to engraftment and development of GvHD ≥ grade 2 in CMV D-R+ patients. 

### 2.3. Data Collection

Pertinent collected data included the following: (i) demographics, (ii) underlying disease and malignancy stage at the time of HCT, (iii) HCT-related variables (conditioning regimen, type of transplant, time to engraftment, D/R CMV serology status, and GvHD: by organ involved, degree of severity, acute versus chronic, and post-HCT day of diagnosis), (iv) CMV-related variables (all specimens tested for CMV, presence of CMV clinical syndrome, date of csCMV infection diagnosis post-HCT, and administered treatments), (v) other dsDNA viral infections, (vi) laboratory data (absolute neutrophil, lymphocyte and platelet counts, renal and liver function variables) on days 7, 84, and 180 of study inclusion, and (vii) outcome variables, including: all-cause mortality, LOS, number of readmissions, and CMV treatment-associated costs.

### 2.4. Institutional Practices 

Per institutional standard operating procedures (SOP), CMVqPCR is performed on plasma once weekly during the first 3 months post-HCT and every other week thereafter until 6 months post-HCT for all CMV R+ patients. Until 16 May 2018, CMV qPCR was performed with the COBAS^®^ AmpliPrep/COBAS^®^ TaqMan^®^ CMV test (Roche Diagnostics, Indianopolis, IN, USA) with a level of detection (LOD) and quantification (LOQ) of 56 and 137 IU/mL, respectively. After 16 May 2018, COBAS^®^ CMV for Cobas^®^ 6800 test (Roche Diagnostics, Indianopolis, IN, USA) was used with a LOD of 21 IU/mL and LOQ of 25 IU/mL. Considering that the study included patients with CMVqPCR measurements before and after this change, for the purposes of this study, the higher LOQ (56 IU/mL) and LOD (137 IU/mL) were used. CMV resistance test was performed by the Clinical Virology-Laboratory Medicine, University Hospital of Basel, Switzerland. Briefly, patients with csCMV infection are treated with foscarnet or (val)ganciclovir, before and after engraftment, respectively, at induction dose until CMVqPCR is non-quantifiable and at maintenance dose for two subsequent weeks.

### 2.5. Definitions

Clinically significant CMV infection was defined based on consensus international guidelines adjusted to our institutional SOP, as any CMV reactivation >150 IU/mL and/or evidence of CMV syndrome/disease requiring initiation of treatment with other-than-letermovir CMV-acting antiviral agent [15,16]. Breakthrough csCMV infection was defined as any infection observed during administration of letermovir. Recurrent csCMV infection was defined as any csCMV infection diagnosed within 7 days after completion of treatment for the prior csCMV infection episode. Study inclusion day was the date of HCT for CMV D-R+ patients and the day of GvHD ≥ grade 2 diagnosis for CMV R+ patients with GvHD. Patients were followed for the first 180 days after study inclusion; there were no patients lost to follow-up. 

### 2.6. Healthcare Resource Utilization Variables

Length of stay was defined as all inpatient days during the first 180 days post-study inclusion. Readmission was defined as any hospitalization for >48 h after discharge from the prior admission. Inpatient CMV treatment-related costs were obtained from the institutional pharmacy records. Public prices for the drugs were extracted from the official Swiss prices (www.listedesspecialites.ch, accessed on 15 September 2020). Hospitalization costs were measured according to the 2018 Swiss accounting REKOLE^®^ system to ensure the accuracy and comparability of costs, 2018 being the most recent year of reference. The 2018 mean cost per patient per hospitalized day in an internal medicine ward was $2568. Costs are expressed in United States (US) $ using the September 2020 monthly average exchange rate (1 CHF = $1.09).

### 2.7. Statistical Analysis

Continuous variables were described as mean and range. Categorical and continuous variables were compared with the Fisher’s exact and a two-tailed Student’s t-test, as appropriate. Cumulative incidence was calculated among cases and controls for the first episode of csCMV infection, censoring for death. Logistic regression was used to identify risk factors for csCMV infection. Independent variables with *p*-value < 0.20 in the univariable analyses were subsequently entered in a backward stepwise fashion into multivariable logistic regression models with mixed effect. Results are presented as odds ratios (OR) with 95% confidence intervals (CI). The overall 6-month mortality was analyzed using Kaplan-Meier survival curves. The log-rank test was used to compare survival distribution between groups. A two-sided test was performed and *p* < 0.05 was considered to be statistically significant. Data were analyzed using STATA 16.0 (StataCorp, College Station, TX, USA) and figures were generated with Graphpad Prism 8.0 (GraphPad Software, Inc., San Diego, CA, USA).

## 3. Results

### 3.1. Patient Population

Seventy-eight patients were included (26 cases and 52 controls), with cases adequately matched to controls (Table 1). Patients were followed for a mean of 163.1 days (range 8, 180), corresponding to 163.1 (range 41, 180) and 163.2 (range 8, 180) days for cases and controls, respectively (*p*-value: 0.99). The proportions of patients who had developed csCMV infection before inclusion were similar in both groups (*p*-value: 1.00).

### 3.2. csCMV Infection 

The cumulative incidence of csCMV infection at day 180 post-study inclusion was 34.6% (9/26) for cases and 82.7% (43/52) for controls (*p*-value < 0.0001; Table 2 and Figure 1A). Incidence rate of csCMV infection was 2.4 and 15.2 per 1000-patient-days for cases and controls, respectively (*p*-value < 0.0001). For patients in the CMV D-R+ group, the cumulative incidence of csCMV infection was 31.3% and 81.3% for cases and controls, respectively (*p*-value: 0.0006; Appendix A, Figure 1B). Incidence rate of csCMV infection was 2.1 and 14.3 per 1000-patient-days for cases and controls, respectively (*p*-value < 0.0001). For patients in the CMV R+/GvHD group, the cumulative incidence of csCMV infection was 40.0% and 85.0% for cases and controls, respectively (*p*-value: 0.01; Appendix A, Figure 1C). Incidence rate of csCMV infection was 2.9 and 10.1 per 1000-patient-days for cases and controls, respectively (*p*-value: 0.0004). 

Thirty-four patients developed more than one csCMV infection: three in cases (11.5%) and 31 in controls (59.6%, *p*-value < 0.0001). Six patients developed CMV biopsy-proven end-organ disease: 1 (3.8%) in cases and 5 (9.6%) in controls (*p*-value: 0.66), involving the central nervous system (three patients), gastrointestinal tract (one patient), and liver (two patients).

Risk factors associated with development of csCMV infection were analyzed in univariable and multivariate regression models (Table 3). The absence of letermovir prophylaxis in controls was the only and strongest predictor of csCMV infection (OR: 9.14, 95%CI 2.94, 28.3, *p*-value < 0.0001).

### 3.3. csCMV Infection Characteristics in Cases

Letermovir was administered for a mean of 98 days (range 4, 258) overall: 97 (range 41, 247) and 100 days (range 4, 200) in the CMV D-R+ and R+/GvHD groups, respectively. In the CMV R+/GvHD group, letermovir was initiated at day 66 post-HCT (mean, range: 10, 175). Among nine cases with csCMV infection, five and four occurred in the CMV D-R+ and CMV R+/GvHD group, respectively. Seven infections were breakthrough csCMV infections, occurring at a mean of 66.6 days (range: 4, 153) after initiation of letermovir prophylaxis: 74.8 days (range: 42, 138) in CMV D-R+ patients versus 55.7 days (range: 4, 153) for CMV R+/GvHD patients (*p*-value: 0.71). Among breakthrough infections, two patients had positive CMVqPCR (118 and 76 IU/mL) at treatment initiation and rapidly developed csCMV infection (on day 5 and 7 post-initiation, respectively). Three patients with breakthrough csCMV infection in the CMV D-R+ group developed acute GvHD preceding csCMV infection, despite continuation of letermovir prophylaxis. There was no letermovir resistance mutation identified in the seven breakthrough csCMV infections. Only two csCMV infections were observed after letermovir prophylaxis discontinuation. 

### 3.4. CMVqPCR Results

Overall, 1773 CMVqPCR tests were performed, 565 (mean: 22/patient; range 6, 37) for cases and 1208 (mean: 23/patient; range 3, 42) for controls (*p*-value: 0.43; Figure 2A). A significantly higher number of negative tests was observed in cases (19.0, range: 6, 34) as compared to controls (11.1, range: 0, 25; *p*-value < 0.0001; Figure 2A,B). Similarly, a significantly lower number of positive/detectable (≥LOD: 56 IU/mL) CMVqPCR (0.9, range: 0, 4) and positive/quantifiable (≥LOQ: 137 IU/mL) CMVqPCR tests (1.8, range: 0, 9) were observed in cases compared to controls (7.0, range: 0, 19; *p*-value < 0.0001 and 6.4, range: 0, 18; *p*-value: 0.007, respectively; Figure 2A,B). Absolute values of all positive/quantifiable CMVqPCR tests were significantly lower in cases (1461 IU/mL, range: 139, 11,900) compared to controls (3727 IU/mL, range: 138, 132,000; *p*-value < 0.0001; Figure 2C). 

### 3.5. CMV-Associated Treatment

Patients with csCMV infection received valganciclovir (68.2%), foscarnet (15.2%), ganciclovir (14.7%), and cidofovir (1.8%) as pre-emptive treatment. Duration of anti-CMV treatments for all episodes of csCMV infections during the study period was significantly lower in cases (49 days) than controls (77.8 days; *p*-value: 0.02; Table 2). The duration of valganciclovir administration was shorter in cases (27.3 days) versus controls (57.1 days; *p*-value: 0.008). Treatment duration with ganciclovir and foscarnet was similar between the two groups. Cidofovir was only administered in three patients, all controls. There was a trend for lower CMV treatment-associated costs in cases ($4096) versus controls ($9736; *p*-value: 0.07). The cost difference was mainly due to costs related to valganciclovir: $1089 and $2281 for cases and controls, respectively (*p*-value: 0.008). The mean cost of letermovir prophylaxis was $38,461 (range 1788, 89,193). 

### 3.6. Other Secondary Outcomes

There was a trend for shorter LOS in cases versus controls (*p*-value: 0.16), with a mean difference of 15 days. Similarly, there was a trend for lower overall hospitalization costs in cases compared to controls (*p*-value: 0.16). There were no significant differences in the proportions of patients requiring a hospital readmission between cases and controls (30.8% and 32.7%, respectively; *p*-value: 1.00). There was a trend for lower hospitalization costs in cases compared to controls (*p*-value: 0.16) but no difference was found in the total costs between cases and controls (*p*-value: 0.75). All-cause mortality at 180 days post-inclusion was 15.4% (4/26) and 25% (13/52) in cases and controls, respectively (*p*-value: 0.40; Appendix A). Thirty patients developed a non-CMV dsDNA viral infection: 12 cases (46.2%) and 21 controls (34.6%, *p*-value: 0.34) (Table 2). 

### 3.7. Hematology and Laboratory Outcomes

No difference in hematologic malignancy relapse rate was observed between cases (15.4%, 4/26) and controls (18.0%, 9/52; *p*-value: 1.00). Among CMV D-R+ patients, time to engraftment was longer in cases (21.2 days) compared to controls (17.4 days; *p*-value: 0.004) and no difference was observed in the incidence of acute GvHD between cases (5/16, 31.3%) and controls (16/32, 50.0 %; *p*-value: 0.36). 

There were no differences in the mean values of neutrophil and lymphocyte counts between cases and controls at any of the predefined time-points (Figure 3A,B). In contrast, significantly higher mean platelet counts were observed by day 84 in cases (134 G/L) compared to controls (73 G/L; *p*-value < 0.001) and by day 180 post-inclusion in cases (138 G/L) compared to controls (92 G/L; *p*-value: 0.03; Figure 3C). When looking at subgroups, platelet counts were significantly increased in cases (132 G/L) compared to controls (79 G/L, *p*-value: 0.01) by day 84 in the D-R+ group (Appendix A). Similarly, in the GvHD group, platelet counts were higher in cases compared to controls (181 G/L vs. 63 G/L, *p*-value < 0.001) by day 180 (Appendix A). Although cases had worse renal function at baseline (GFR: 76 mL/min/1.73 m^2^) compared to controls (GFR: 92 mL/min/1.73 m^2^; *p*-value: 0.04), by the end of the study, GFR was significantly improved in cases (94 mL/min/1.73 m^2^) versus controls (74 mL/min/1.73 m^2^; *p*-value: 0.006; Figure 3D). There were no differences observed in liver function tests between cases and controls.

## 4. Discussion

With an excellent safety profile and efficacy, letermovir constitutes a promising novel anti-CMV agent that harbors the longstanding hope of efficiently preventing clinical and economic burden of CMV in immunocompromised patients. Using a retrospective matched cohort approach, our results confirm the drastic reduction of csCMV infections in letermovir-treated patients, but also give unique insights on related resource utilization and hematologic outcomes.

Administration of letermovir significantly reduced the incidence of csCMV infections, in line with previous observations [11,12,13]. The relatively higher rates of csCMV infections observed in our study could, in part, be attributed to the low cutoff of CMVqPCR, above which csCMV infection was considered, and the high-risk patient population, slightly differing from the pivotal clinical trial inclusion criteria [11,12,14]. The intravenous formulation of letermovir was not available during the study period; hence, only PO letermovir was used. Considering the frequency of gastrointestinal mucositis and GvHD, suboptimal letermovir absorption could have contributed to breakthrough csCMV infections. Notably, in all breakthrough infections, no resistance-conferring mutation within the UL56-CMV-terminase gene was identified. Two patients developed a breakthrough csCMV infection within the first week of letermovir initiation, both with already detectable CMV viral loads at the time of their inclusion. This is consistent with post-hoc data of the letermovir registration trial, showing higher rates of csCMV infection in patients with detectable versus undetectable CMV-viral loads, and with our recent report on risk factors for breakthrough csCMV infection during letermovir prophylaxis [17,18]. Regarding breakthrough csCMV infection, it is worth noting that our strict threshold for csCMV infection (150 IU/mL) might have underestimated letermovir efficacy and led to unnecessary preemptive treatment use and costs. Thus, due to the unique mechanism of the action of letermovir, which prevents infectious virions production but not DNA synthesis, the clinical relevance of low-grade CMV viremia remains to be clarified. 

The administration of letermovir was associated with shorter overall duration and lower costs of CMV-associated treatments. This was mainly due to a reduction in the duration of valganciclovir administration, frequently used for the management of csCMV infection on an outpatient basis. As controls were more likely to have >1 episode of csCMV infection, these findings suggest that although the duration of intravenously administered CMV-associated treatments and rates of readmission were similar between cases and controls, administration of letermovir might have an important impact on the outpatient management of csCMV infections. Due to the retrospective nature of the study, we were not able to assess the overall burden of CMV infection on the outpatient setting. However, as csCMV infection post-HCT leads to high numbers of outpatient visits, costs, efforts, and significant stress for patients and healthcare personnel, the benefits of letermovir prophylaxis may extend beyond hospitalized patients and decrease the ambulatory burden. 

The cost of letermovir prophylaxis accounted for >90% of CMV-related costs in cases. However, and despite the relatively short follow-up and small number of patients included in our study, we could still observe a trend for lower CMV treatment-associated costs, overall hospitalization costs and shorter LOS in cases compared to controls. Calculating the total (hospitalization and anti-CMV drugs) costs, the letermovir cost was absorbed by the relatively lower hospitalization costs in cases, as no difference could be observed between the two groups. Our cost and healthcare resources analysis is however limited by a number of reasons, including the fact that hospitalization costs were measured according to the 2018 Swiss accounting REKOLE^®^ system and does not reflect patient-specific associated costs. In addition, outpatient CMV-associated resource utilization and indirect costs associated with preemptive anti-CMV treatments, which have been previously well-described post-HCT, were not taken into consideration [2,8]. In addition, >1 csCMV infection was also reported to increase the overall costs of an allogeneic HCT by 25–30% [2]. Clearly, more data are required to better describe the direct and indirect costs and benefits associated with letermovir use in high-risk hematology patients. 

Our study is the first to indicate a hematological benefit in patients treated with letermovir, with more robust platelet count recovery observed. This improvement could be attributed to less csCMV infection recurrences and overall exposure to valganciclovir treatment, two common causes of cytopenias related to poor graft function [19]. Given the direct and indirect effects of CMV infection on bone marrow function and the sustained impairment of hematopoietic stem cell self-renewal and proliferation during this chronic inflammatory state, letermovir-treated patients may be at lower risk for associated cytopenias [20]. Although no differences in neutrophil and lymphocyte counts were observed between cases and controls, platelet recovery may be a better marker of graft function, due to the absence of routinely used stimulating factor as G-CSF. Regarding anti-CMV immune function, the observed improvement on immune reconstitution should be balanced by a potential delay in CMV-specific T-cell reconstitution due to a reduced antigen exposure with antiviral prophylaxis. Similarly to ganciclovir, letermovir prophylaxis has been recently associated to such a delayed polyfunctional CMV-specific cellular immune reconstitution, although the impact on late csCMV infection needs to be evaluated on larger studies [21,22,23]. Finally, we also report an effect of letermovir on renal function, which might also be explained by lower exposure to anti-CMV drugs toxicities. We did not find any difference in non-CMV viral infections incidence between cases and controls.

Finally, our study is limited by its retrospective nature, in addition to the small population size. Due to its real-world observational design, ambulatory follow-up and hospitalizations in other centers could not be studied in detail. Cost analysis was also complicated by the inherent delay due to annual financial reports generation. In conclusion, our data confirm the efficacy of letermovir primary prophylaxis in preventing csCMV infection in a real-world setting. We report significantly shorter treatment courses of treatment with valganciclovir, suggesting a significant effect on the outpatient management of CMV infections, with trends for shorter overall hospital LOS. Robust platelet count recovery in patients treated with letermovir requires further studies. 

## Figures and Tables

**Figure 1 vaccines-09-00372-f001:**
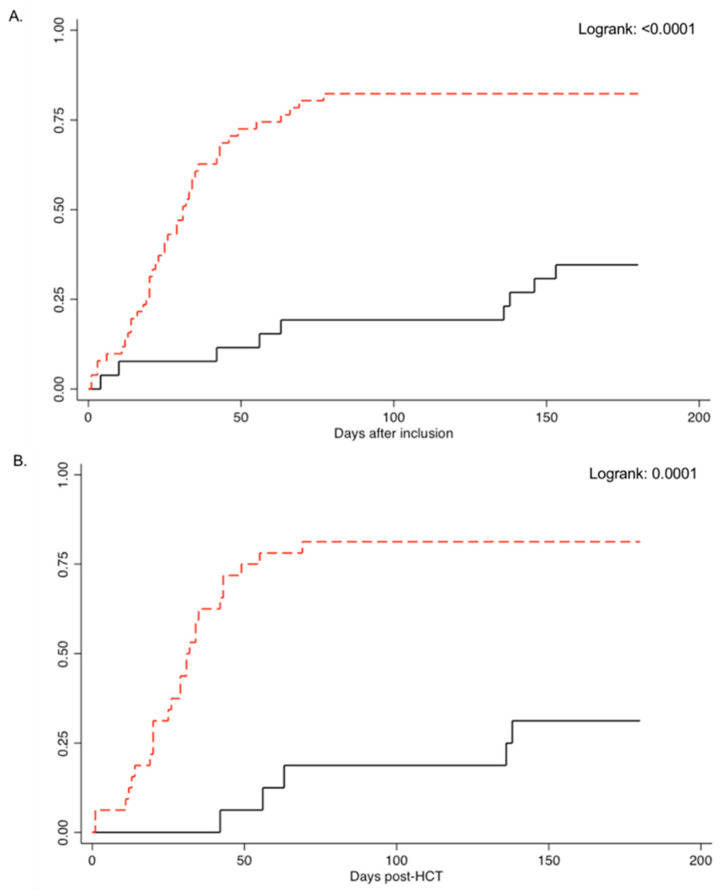
Cumulative incidence of clinically significant CMV infection between cases (black line) and controls (red line) in the (**A**) overall patient population, (**B**) CMV donor negative/recipient positive group, and (**C**) CMV recipient positive with graft versus host disease group.

**Figure 2 vaccines-09-00372-f002:**
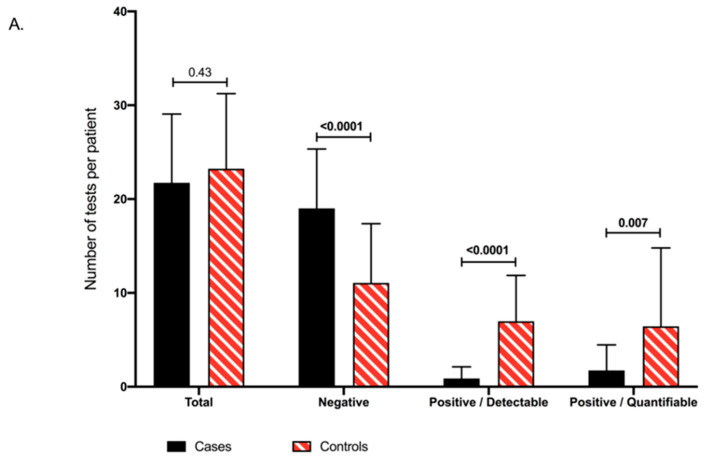
(**A**) Numbers of quantitative CMV PCR detection tests performed during the study period between cases and controls, presented as negative (<56 IU/mL), positive/detectable (56–136 IU/mL) and positive/quantifiable (>137 IU/mL). (**B**) Proportions of negative, positive/detectable and positive/quantifiable CMV PCR tests between cases and controls. (**C**) Absolute values of positive/quantifiable quantitative CMV PCR tests between cases and controls.

**Figure 3 vaccines-09-00372-f003:**
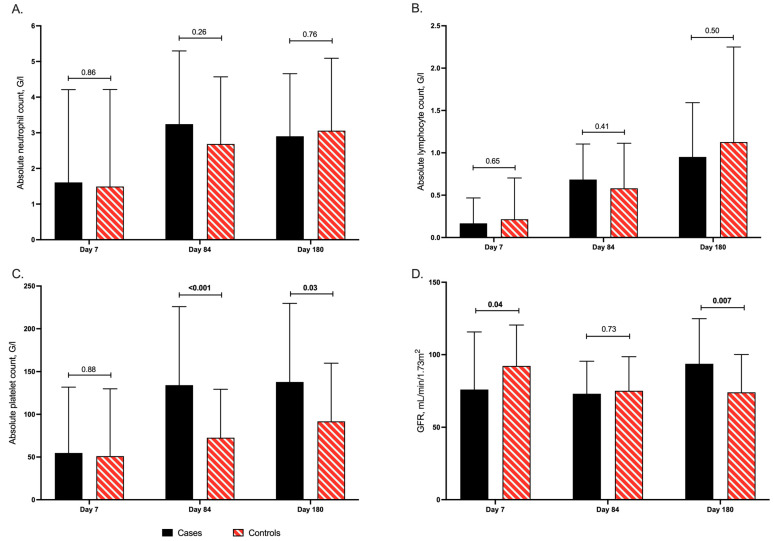
Distribution of (**A**) absolute neutrophil (G/L), (**B**) absolute lymphocyte (G/L), (**C**) platelet counts (G/L), and (**D**) glomerular filtration rate (mL/min/1. 73 m^2^) by days 7, 84, and 180 after study inclusion in the overall study patient population.

**Table 1 vaccines-09-00372-t001:** Baseline patient characteristics.

Patient and HCT Characteristics	All Patients *n* = 78 (%)	Cases *n* = 26 (%)	Controls *n* = 52 (%)	*p*-Value
Demographics				
Age (Years), Mean (Range)	55.3 (17, 74)	55.8 (17, 73)	55.0 (21, 74)	0.81
Gender, Female	30 (38.5)	9 (34.6)	21 (40.4)	0.81
Underlying disease				
Myeloid hematologic malignancy	56 (71.8)	18 (69.2)	38 (73.1)	0.79
Lymphoid hematologic malignancy	22 (28.2)	8 (30.8)	14 (26.9)	
Malignancy remission before HCT	67 (85.9)	21 (80.8)	46 (88.5)	0.49
HCT Characteristics				
Conditioning, Myeloablative	19 (24.4)	7 (26.9)	12 (23.1)	0.78
HCT donor-recipient matching				0.95
HLA-matched related	13 (16.7)	4 (15.4)	9 (17.3)	
HLA-matched unrelated	42 (53.9)	15 (57.7)	27 (51.9)	
Haploidentical	23 (29.5)	7 (26.9)	16 (30.8)	
HCT source				0.53
Bone marrow	13 (16.7)	3 (11.5)	10 (19.2)	
Peripheral blood	65 (83.3)	23 (88.5)	42 (80.8)	
GvHD ^1^				
GvHD grade ≥ 2	30 (38.5)	10 (38.5)	20 (38.5)	1.00
Acute GvHD	30 (38.5)	10 (38.5)	20 (38.5)	1.00
Refractory GvHD	2 (2.6)	1 (3.9)	1 (1.9)	1.00
GIT GvHD	22 (28.2)	8 (30.8)	14 (26.9)	0.79
Serologies				
CMV serological status				1.00
Donor+/Recipient+	27 (34.6)	9 (34.6)	18 (34.6)	
Donor−/Recipient+	51 (65.4)	17 (65.4)	34 (65.4)	
EBV serological status				0.56
Donor+/Recipient−	4 (5.1)	2 (7.7)	2 (3.9)	
Donor+/Recipient+	68 (87.2)	23 (88.5)	45 (86.5)	
Donor−/Recipient+	6 (7.7)	1 (3.9)	5 (9.6)	
Toxoplasmosis serological status				0.71
Donor−/Recipient−	22 (28.2)	8 (30.8)	14 (26.9)	
Donor+/Recipient−	4 (5.1)	1 (3.9)	3 (5.8)	
Donor+/Recipient+	19 (24.4)	8 (30.8)	11 (21.2)	
Donor−/Recipient+	33 (42.3)	9 (34.6)	24 (46.2)	
csCMV infection prior to study inclusion	3 (3.9)	1 (3.9)	2 (3.9)	1.00

HCT: Hematopoietic Cell Transplant, HLA: Human Leukocyte Antigen, GvHD: Graft versus Host Disease, GIT: Gastro-Intestinal, CMV: Cytomegalovirus, EBV: Epstein Barr Virus, csCMV: Clinically Significant CMV. ^1^ Information on GvHD was recorded at study inclusion.

**Table 2 vaccines-09-00372-t002:** Primary and secondary clinical outcomes between cases and controls during the first 180 days post-study inclusion ^1^.

	Cases *n*: 26 (%)	Controls *n*: 52 (%)	*p*-Value
**Primary outcome ^1^**			
csCMV infection ^2^	9 (34.6)	43 (82.7)	<0.0001
**Secondary outcomes** ^1^			
**>1 csCMV infection ^3^**	3 (11.5)	31 (59.6)	<0.0001
**CMV treatment duration** ^4^	49 (15, 104)	77.8 (8, 155)	0.02
Ganciclovir	24 (14, 34)	24 (5, 67)	1.00
Valganciclovir	27.3 (8, 70)	57.1 (14, 142)	0.008
Foscarnet	23.5 (5, 41)	20.7 (4, 84)	0.79
Cidofovir		22.3 (8, 31)	
**CMV treatment costs** ^5^	3758 (550, 10,115)	8932 (770, 32,121)	0.07
Ganciclovir	4155 (1545, 6465)	3145 (71, 8188)	0.60
Valganciclovir	999 (293, 2565)	2093 (513, 5204)	0.008
Foscarnet	5842 (1384, 9455)	9454 (1614, 29,519)	0.43
Cidofovir		4931 (2336, 6229)	
**Letermovir costs** ^5^	38,461 (1788, 89,193)	NA	
**Length of stay** ^6^	44.8 (2, 109)	59.8 (3, 180)	0.16
**Readmission**	8 (30.8)	17 (32.7)	1.00
>1 Readmission	2 (7.7)	6 (11.5)	0.71
**Hospitalization costs** ^7^	115,025 (5136, 279,912)	153,370 (7704, 462,240)	0.16
**Total costs** ^8^	142,763 (3106, 348,957)	151,849 (1356, 488,389)	0.75
**All-cause 6-month mortality**	4 (15.4)	13 (25.0)	0.40
**Non-CMV viral infection** ^9^	12 (46.2)	18 (34.6)	0.34
Herpes simplex virus 1/2	0	3 (5.8)	0.55
Epstein-Barr virus	6 (23.1)	5 (9.6)	0.17
Human herpes virus 6	1 (3.9)	4 (7.7)	0.66
Adenovirus	0	4 (7.7)	0.30
BK-virus	5 (19.2)	5 (9.6)	0.29

csCMV: Clinically Significant Cytomegalovirus, D-: Donor Negative, R+: Recipient Positive, GvHD: Graft versus host Disease, HCT: Hematopoietic Cell Transplant, GvHD: Graft versus Host Disease, NA: Not Applicable. ^1^ Results are presented from study inclusion and up to day 180 post-study inclusion. Numerical variables are presented as mean (range). For patients in the CMV donor negative/recipient positive group, day of study inclusion coincided with HCT day. ^2^ The first documented episode of csCMV infection and or disease, for patients who had >1 episode. Seven of nine case-patients tested for letermovir-resistance were negative. There was only one case of CMV disease in one control-patient. ^3^ There were three case-patients who had two csCMV infection episodes. There were 26 and 5 control-patients with two and three csCMV infection episodes, respectively. ^4^ Treatment duration represents CMV pre-emptive and targeted treatment that was administered for all documented episodes of csCMV infections/disease during the study period (from study inclusion and up to day 180 post-study inclusion). Results are presented as mean days (range). ^5^ Costs are presented in US$. Results are presented as mean (range). ^6^ Length of stay refers to the overall length of stay from study inclusion until day 180 post-study inclusion. For patients with >1 admission, the length of stay was calculated by adding all days of hospitalization during the study period. Results are presented as mean days (range). ^7^ Hospitalization costs were measured according to the 2018 Swiss accounting REKOLE^®^ system, 2018 being the most recent year of reference. The 2018 mean cost per patient per hospitalized day in an internal medicine ward was US $2568. Estimated total hospitalization costs were calculated by multiplying the length of stay for each patient in days by US $2568. ^8^ Total costs were calculated by adding all anti-CMV drugs costs (including letermovir) and hospitalization costs during the study period. Results are presented as mean days (range). ^9^ Two control-patients in the CMV D-R+ group had >1 non-CMV double-stranded DNA viral infection.

**Table 3 vaccines-09-00372-t003:** Risk factors analysis for clinically significant CMV infection.

Variables	Univariable Analysis			Multivariable Analysis		
	Odds Ratio	95% CI	*p*-Value	Odds Ratio	95% CI	*p*-Value
**Demographics**						
Age (Years), Mean (Range)	1.01	0.98, 1.04	0.41			
Gender, Female vs. Male	0.78	0.29, 2.08	0.62			
**Underlying disease**						
Myeloid vs. Lymphoid hematologic malignancy	1.10	0.38, 3.15	0.86			
Malignancy remission before HCT, Yes vs. No	0.35	0.10, 1.30	0.12	0.39	0.09, 1.70	0.21
**HCT Characteristics**						
Conditioning, Non-Myeloablative vs. Myeloablative	0.45	0.15, 1.30	0.14	0.47	0.13, 1.66	0.24
HLA-matched related vs. HLA-matched unrelated vs. Haploidentical	1.55	0.75, 3.17	0.23			
Bone marrow vs. Peripheral blood stem cells	3.22	0.66, 15.77	0.15	2.44	0.38, 15.70	0.35
GvHD grade ≥ 2 at baseline, Yes vs. No	1.28	0.48, 3.41	0.62			
GvHD grade ≥2 post baseline, Yes vs. No	2.22	0.72, 6.89	0.17			
Acute GvHD grade ≥ 2 at baseline, Yes vs. No	1.28	0.48, 3.41	0.62			
Acute GvHD grade ≥2 post baseline, Yes vs. No	1.87	0.60, 5.83	0.28			
Refractory GvHD at baseline, Yes vs. No	0.49	0.03, 8.17	0.62			
GIT GvHD at baseline, Yes vs. No	1.48	0.50, 4.39	0.48			
**Serologies**						
CMV serological status, D + R+ vs. D-R+	0.77	0.28, 2.11	0.61			
EBV serological status	1.17	0.44, 3.14	0.75			
Toxoplasmosis serological status	1.08	0.71, 1.64	0.72			
**Controls vs. Cases**	9.02	3.06, 26.61	<0.0001	9.14	2.94, 28.3	<0.0001

HCT: Hematopoietic Cell Transplant, GvHD: GvHD: Graft versus Host Disease, D: Donor, R: Recipient, +: Positive, −: Negative, GIT: gastro-intestinal tract. Only variables with a *p*-value ≤ 0.15 were entered in a stepwise fashion into the multivariable model. The Hosmer-Lemeshow chi2 value was 3.8, with a probability of 0.44, suggesting a good fit for this model.

## Data Availability

Data are available on request from the authors.

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
