# Peer review of "Letermovir Primary Prophylaxis in High-Risk Hematopoietic Cell Transplant Recipients: A Matched Cohort Study"

_vaccines, 2021, doi:10.3390/vaccines9040372_

Round 1

Reviewer 1 Report

Authors evaluated the effect of letermovir prophylaxis for CMV infection during 18 months after hematopoietic cell transplantation retrospectively. Authors showed that the treatment could reduce of CMV infection significantly, improve allograft engraftment efficacy shown by increasing PLT number and maintain renal function better than the control group. The number of cases is limited and the follow up period relatively short (18 months), which could be part of the reason that mortality and malignant tumor development did not show significant differences. 

The report is interesting and showing clinically important information.

There are some information that should be added or corrected.

  • In patient population, there are two groups of letermovir treatment 1) Donor CMV(-) Recipient CMV (+) receiving letemorvir p.o. for 100 days, and 2) allograft CMV (+) and showing early stage 2 GVHD+ , treated with letermovir under steroid treatment. Between the two groups, the total dose of drug and duration of treatment may differ, therefore, it would be more informative to analyze engraftment efficacy and renal function separately in these two groups as shown in supplemental Table. This information could be added as supplemental figures and mentioned in the results.

  • In Supplemental table,

Letermovir

37257 (2336, 6229)

NA

32129 (1640, 64553)

NA

 Is this cost of treatment? In the left number 37257 (2336, 6229) indicates mean (minimal value, max value)? If it is so, the number must be wrong. Please correct the table.

  • To discuss the costs efficacy of letermovir treatment, total sum of medication cost after hematopoietic cell transplantation could be compared between letermovir treated group and control group, i.e. cost of letermovir should be included as part of treatment cost. It is interesting to know whether the treatment with letermovir significantly lower the CMV infection episodes, which should reduce total cost of treatment significantly.

  • Ganciclovir prophylaxis reduce the risk of CMV infection and disease after hematopoietic cell transplantation. However, it is also known to delay the memory cell development against CMV after transplantation, thus increasing late onset CMV infection. Whether the letermovir primary prophylaxis can lead to better development of immunity against CMV, compared to ganciclovir and make recipients more resistant against CMV infection is an important point. Please add this possibility in the discussion.

Author Response

Manuscript ID vaccines-1170449: Letermovir primary prophylaxis in high-risk hematopoietic cell transplant recipients: a matched cohort study

Comment of the authors: We thank the reviewers for considering our work and for giving us the opportunity to revise it. Please find hereby a point-by-point response to the reviewer’s comments and a revision of the manuscript including the requested modifications.

Reviewer 1

Authors evaluated the effect of letermovir prophylaxis for CMV infection during 18 months after hematopoietic cell transplantation retrospectively. Authors showed that the treatment could reduce of CMV infection significantly, improve allograft engraftment efficacy shown by increasing PLT number and maintain renal function better than the control group. The number of cases is limited and the follow up period relatively short (18 months), which could be part of the reason that mortality and malignant tumor development did not show significant differences. 

The report is interesting and showing clinically important information.

There are some information that should be added or corrected.

  • In patient population, there are two groups of letermovir treatment 1) Donor CMV(-) Recipient CMV (+) receiving letermovir p.o. for 100 days, and 2) allograft CMV (+) and showing early stage 2 GVHD+ , treated with letermovir under steroid treatment. Between the two groups, the total dose of drug and duration of treatment may differ, therefore, it would be more informative to analyze engraftment efficacy and renal function separately in these two groups as shown in supplemental Table. This information could be added as supplemental figures and mentioned in the results.

Response:

We thank the reviewer for his interest in our work and for raising interesting points. To further analyze platelet recovery and renal function, we added, as suggested, a supplementary figure (Figure 2 supplementary material) with subgroup analysis of these two parameters. Interestingly, the timing differed between subgroups, with a significant increase in platelet counts in cases compared to controls at day 84 among the D-R+ subgroup, while at day 180 among the GvHD group. A sentence was added subsequently in the Result section (lines 247-250).

  • In Supplemental table,

Letermovir

37257 (2336, 6229)

NA

32129 (1640, 64553)

NA

 Is this cost of treatment? In the left number 37257 (2336, 6229) indicates mean (minimal value, max value)? If it is so, the number must be wrong. Please correct the table.

Response:

We thank the reviewer for his careful reading and corrected the incorrect numbers in Supplement Table 1.

  • To discuss the costs efficacy of letermovir treatment, total sum of medication cost after hematopoietic cell transplantation could be compared between letermovir treated group and control group, i.e. cost of letermovir should be included as part of treatment cost. It is interesting to know whether the treatment with letermovir significantly lower the CMV infection episodes, which should reduce total cost of treatment significantly.

Response:

We agree that a global view on costs of letermovir in allogeneic hematopoietic transplant recipients would be interesting and needs to be added to the text. We added three lines in Table 2 including “Letermovir costs”, “Hospitalization costs” and “Total costs” (calculated as the sum of hospitalization costs and anti-CMV drugs costs including letermovir). No significant difference was observed in total costs during the study period between cases and controls. This is largely due to the expensive price of letermovir prophylaxis (mean of $38’461) in Switzerland, as both CMV treatments and hospitalization costs were reduced in letermovir recipients compared to controls. A few sentences were added in the Result section (lines 231-233) and in the Discussion section (lines 300-310).

  • Ganciclovir prophylaxis reduce the risk of CMV infection and disease after hematopoietic cell transplantation. However, it is also known to delay the memory cell development against CMV after transplantation, thus increasing late onset CMV infection. Whether the letermovir primary prophylaxis can lead to better development of immunity against CMV, compared to ganciclovir and make recipients more resistant against CMV infection is an important point. Please add this possibility in the discussion.

Response:

We thank the reviewer for raising this interesting point that was not discussed in our manuscript. As shown with ganciclovir, two recent articles suggest that letermovir could delay the CMV-specific cellular immune reconstitution, due to a reduced antigen exposure. The improvement on hematologic reconstitution that we observe in the present study might mitigate the impact of the reduction in specific cellular immune reconstitution, however the risk of late csCMV infection in letermovir recipients remains to be defined in large studies. We added few sentences and recent references in the Discussion section (lines 322-329).

Reviewer 2 Report

I was invited to revise the paper entitled "Letermovir primary prophylaxis in high-risk hematopoietic cell transplant recipients: a matched cohort study". It was a cohhort study aimed to evaluate the impact of Letermovir prophylaxis on CMV infection among hematopoietic transplanted patients. Authors evaluated both survival to CMV infection and associated factors.

I want to congratulate with Authors for the excellent work and for the outstanding results.

Introduction was well written and deeply described the study background.

Methods are adequate and well described. Results are clear and easy to read for the reader. 

Conclusions are supported by results.

I have only some minor comments:

  • I suggest to perform Cox regression to evaluate risk factors;
  • I suggest to discuss about cost analysis that was not mentioned in "Discussion" section apart of limitations.

Author Response

Manuscript ID vaccines-1170449: Letermovir primary prophylaxis in high-risk hematopoietic cell transplant recipients: a matched cohort study

Comment of the authors: We thank the reviewers for considering our work and for giving us the opportunity to revise it. Please find hereby a point-by-point response to the reviewer’s comments and a revision of the manuscript including the requested modifications.

Reviewer 2:

I was invited to revise the paper entitled "Letermovir primary prophylaxis in high-risk hematopoietic cell transplant recipients: a matched cohort study". It was a cohort study aimed to evaluate the impact of Letermovir prophylaxis on CMV infection among hematopoietic transplanted patients. Authors evaluated both survival to CMV infection and associated factors.

I want to congratulate with Authors for the excellent work and for the outstanding results.

Introduction was well written and deeply described the study background.

Methods are adequate and well described. Results are clear and easy to read for the reader. 

Conclusions are supported by results.

I have only some minor comments:

  • I suggest to perform Cox regression to evaluate risk factors;

Response:

We thank the reviewer for his kind and favorable comments on our work. We do agree that cox regression is more appropriate approach for cohort studies. However, as this was a matched cohort study with 1:2 case:control ratio, we deemed more appropriate to perform logistic regression.

  • I suggest to discuss about cost analysis that was not mentioned in "Discussion" section apart of limitations.

Response:

We thank the reviewer for his interesting suggestion. Three lines were added in Table 2 including “Letermovir costs”, “Hospitalization costs” and “Total costs” (calculated as the sum of hospitalization costs and anti-CMV drugs costs including letermovir). No significant difference was observed in total costs during the study period between cases and controls. This is largely due to the expensive price of letermovir prophylaxis (mean of $38’461) in Switzerland, as both CMV treatments and hospitalization costs were reduced in letermovir recipients compared to controls. A few sentences were added in the Result section (lines 231-233) and in the Discussion section (lines 300-310).

Reviewer 3 Report

The design of the study is adequate. The conclusions are appropriate and consistent with design of the study and results.

No relevant issues are observed.

Only minor issues:

  • table 1: line acute GVHD is redundant (since grade>= 2 GVHD is reported);
  • in the Discussion the Authors should comment the results of the non-CMV infection results they reported in the results. 

Author Response

Manuscript ID vaccines-1170449: Letermovir primary prophylaxis in high-risk hematopoietic cell transplant recipients: a matched cohort study

Comment of the authors: We thank the reviewers for considering our work and for giving us the opportunity to revise it. Please find hereby a point-by-point response to the reviewer’s comments and a revision of the manuscript including the requested modifications.

Reviewer 3

The design of the study is adequate. The conclusions are appropriate and consistent with design of the study and results.

No relevant issues are observed.

Only minor issues:

  • table 1: line acute GVHD is redundant (since grade>= 2 GVHD is reported);

Response:

We thank the reviewer for his positive remarks on our work.

Regarding his remark on Table 1, although numbers are similar between lines “GvHD ≥ grade2” and “acute GvHD”, the first one specifically refers to the time of onset (acute vs chronic GvHD) and the second to the grade of the disease (1 vs ≥2 grade GvHD). Among GvHD ≥ grade 2, all of them were diagnosed as acute GvHD and no chronic GvHD was recorded.

  • in the Discussion the Authors should comment the results of the non-CMV infection results they reported in the results.

Response:

We thank the reviewer for this interesting suggestion and added a sentence on non-CMV viral infections in the Discussion section (lines 328-329).
